# Probiotic Supplementation and High-Intensity Interval Training Modify Anxiety-Like Behaviors and Corticosterone in High-Fat Diet-Induced Obesity Mice

**DOI:** 10.3390/nu13061762

**Published:** 2021-05-21

**Authors:** Parisa Foroozan, Maryam Koushkie Jahromi, Javad Nemati, Hosein Sepehri, Mohammad Amin Safari, Serge Brand

**Affiliations:** 1Department of Sport Sciences, School of Education and Psychology, Shiraz University, Shiraz 7194684334, Iran; foroozan_parisa57@yahoo.com (P.F.); koushkie53@yahoo.com (M.K.J.); nemati_phy@yahoo.com (J.N.); hosseinsepehri63@gmail.com (H.S.); aminsafari70@yahoo.com (M.A.S.); 2Center for Affective, Stress and Sleep Disorders (ZASS), Psychiatric University Hospital Basel, 4002 Basel, Switzerland; 3Department of Clinical Research, University of Basel, 4031 Basel, Switzerland; 4Substance Abuse Prevention Research Center, Health Institute, Kermanshah University of Medical Sciences, Kermanshah 6714869914, Iran; 5Sleep Disorders Research Center, Kermanshah University of Medical Sciences, Kermanshah 6714869914, Iran; 6Department of Sport, Exercise and Health, Division of Sport Science and Psychosocial Health, University of Basel, 4052 Basel, Switzerland; 7School of Medicine, Tehran University of Medical Sciences, Tehran 1417613151, Iran

**Keywords:** probiotics, diet, high-fat, high-intensity interval training, anxiety, corticosterone

## Abstract

Evidence supports the role of exercise training and probiotics on reducing obesity. Considering the relationship between obesity and high-fat diet with anxiety indices, the aim of this study was to assess the effect of probiotic supplementation and high-intensity interval training (HIIT) on anxiety-like behaviors, corticosterone and obesity indices in high-fat diet (HFD)-induced obesity mice. Thirty male adult C57BL/6 mice were randomly divided into five groups: (1) Control with normal diet (CON), (2) High-fat diet (HFD), (3) HFD + exercise training (HT), (4) HFD + probiotics supplement (HP) and (5) HFD + exercise training +probiotics (HTP). Exercise training consisted of 8 weeks of high-intensity interval training (HIIT) programs. Probiotics supplement included 0.2 mL *Lactobacillus rhamnosus GG*. Anxiety-like behaviors were measured by open field (OF) and Elevated plus maze (EPM). OF and EPM tests, visceral fat mass (VFM) measurement, and blood sampling for corticosterone were performed after the intervention. Bodyweight was measured at different stages during the intervention. HFD regime in C57BL/6 mice increased bodyweight, VFM, and serum corticosterone levels and anxiety-like behaviors (*p* < 0.05). HIIT, probiotic and their combination, decreased bodyweight, VFM, and serum corticosterone levels and improved anxiety-like behavior in the HFD mice (*p* < 0.05). The effect of a combination of HIIT and probiotic on most of the anxiety indices was more than each one separately (*p* < 0.5). HIIT and probiotic supplements separately or above all in combination, may have beneficial effects in reducing obesity and anxiety indices.

## 1. Introduction

Anxiety is a common psychiatric disorder that is marked by excessive fear, fatigue, heart palpitations and tension [1]. Unhealthy human diets are associated with cognitive and affective disorders, including anxiety [2,3]. Further, long-term consumption of a high-fat diet (HFD) increases anxiety-like behaviors in both human and animals [4]. Although the underlying mechanism of the relationship between HFD feeding and anxiety is unclear [4], HFD increased body fat accumulation and metabolic disorders, impaired hypothalamic pituitary adrenal axis (HPA), and increased circulating corticosterone levels. Excessive corticosterone secretion was associated with increased systemic inflammation and behavioral disorders [4]. HFD causes negative effects on neurovascular coupling and cerebrovascular function even without presence of dyslipidemia [5] that may be related to anxiety. However, the most effective intervention to treat or prevent HFD-induced anxiety is not clear so far.

Recent studies showed that exercise can reduce some of the problems associated with HFD, such as anxiety, and improve HPA-axis activity [6]. Further, high-intensity interval training (HIIT) appears to have a more profound influence on metabolic function and aerobic capacity than low and moderate-intensity training in human and rodent studies [7]. HIIT can decrease fat mass, enhance insulin sensitivity, and improve muscle metabolic adaptations. This suggests that this type of exercise induces a positive impact on corticosterone status [7] and may improve the functional state of the HPA axis in HFD treated rats. However, it is surprising that no study was found to explore interactions between HIIT with HFD on anxiety behaviors and corticosterone responses in obese subjects. Indeed, HIIT and HFD can interact through various physiological mechanisms such as gut microbiota.

Gut microbial composition may influence brain health through improving brain development and function [8]. The gut microbiota can be improved or restored through live microorganism probiotic supplementation that may have health benefits [9]. Meanwhile, unhealthy HFD causes marked changes in the gut microbiota in mice [10]. Probiotic supplementation (PS) may control weight gain and the problems associated with HFD in hamsters [11] and reduce body weight and fat mass through several mechanisms such as decreasing insulin resistance and increasing satiety [12]. Probiotics can also ameliorate many side effects of the HFD state and reduce anxiety via the activation of neural pathways [11]. A recent report demonstrated that anxiety-like behaviors induced by chronic mild stress reduced by the probiotic administration in mice [13]. In addition, probiotic supplementation (PS) can reduce serum corticosterone level in mice [14]. Therefore, PS may have a positive impact on the management of psychological disorders and can be considered as a valuable therapeutic intervention for treating anxiety disorders-related HFD state. One of the widely used probiotics are *Lactobacillus rhamnosus GG* that can have a greater and longer lasting effect due to its resistance to acid and bile, good growth and adhesion capability to the intestinal epithelial layer [15]. Also, anti-obesity properties of *Lactobacillus rhamnosus GG* [16] even in HFD conditions [17] has been indicated in some previous studies. Considering the relationship between HFD and anxiety, the aim of this study was to examine the effect of probiotic as *Lactobacillus rhamnosus GG* supplementation, separately or combined with HIIT, on the anxiety-like behaviors and corticosterone in high-fat diet-induced obese mice.

## 2. Materials and Methods

### 2.1. Animals, Diets and Supplement

The study procedures were approved by the local ethic and graduate study committee of Shiraz University (registration no. 2,632,055 at 17 June 2020) and were in accordance with the Helsinki Treat (Laboratory Animal Ethics).

Considering the statistical test and number of groups, the number of animals in each group was calculated as five [18] and for considering any reduction of samples, six rats were allocated to each group. So, 30 adult male C57BL/6 mice (7 ± 1 weeks) weighing 20.7 ± 1 g were provided from the institute of laboratory animal breeding. The mice housed two-per-cage in an animal lab under standard conditions (12 h light/dark cycle in a room at a temperature of 20–25 °C) with access to food and water ad libitum. A normal diet (*n* = 6) consisted of a total of 3.96 kcal/g, carbohydrate 66.40%, fat 10.60% and protein 23%, and HFD (*n* = 24) contained 4.73 kcals/g, 45% of fat, 20% of protein, and 35% of carbohydrate. A high-fat diet continued for all groups except for CON for 18 weeks. For energy intake evaluation, the food consumption of animals was measured at the end of every evening, immediately following the dark cycle.

### 2.2. Inducing Obesity and Differentiation of Groups

In order to make the mice obese, the C57BL/6 mice were subjected to HFD regime for 8 weeks, and showed a significant increase in weight compared to CON group of mice (*p* < 0.001; Figure 1A). The animals were divided randomly into five groups: (1) normal diet as control group (CON, *n* = 6), (2) high-fat diet (HFD, *n* = 6), (3) HFD plus HIIT (HT, *n* = 6), (4) HFD plus probiotics (HP, *n* = 6) and (5) HFD plus HIIT and probiotics (HTP, *n* = 6). Interventions (HIIT and/PS) continued for 8 weeks in HFD mice. HP and HTP groups of C57BL/6 mice received probiotics supplement. Groups not receiving supplements were gavaged with phosphate-buffered saline (PBS). C57BL/6 mice body weights were measured at the same time every day during the study by a digital balance (model 707; Seca, Hamburg, Germany). As indicated in the Figure 1A, the weight of HFD mice show continuous increase until the last week of experiment, so that 5 groups of mice with HFD had more bodyweight (30.02 ± 3.52 g) compared to the CON group (28.90 ± 3.54 g) (*p* < 0.001).

### 2.3. Probiotic Supplement

*Lactobacillus rhamnosus GG* supplement was prepared as a solution from Pardis Roshd Mehregan Bioproducts Company (Parsi Lact, Shiraz, Iran). By insulin syringe and gavage needle, 0.2 mL of bacterial solution was injected daily into each rat’s stomach by gavage. The daily dose of the supplement was 108 × 1 colony forming unit (CFU) per mouse. Groups that did not receive the supplement were gavaged with 0.2 mL of PBS.

### 2.4. Treadmill Training Protocol

For familiarization with the exercise training environment, prior to the main exercise intervention, mice were placed on the stationary treadmill 5 days a week for 1 week. The program of placing on a treadmill continued until the end of experiment for 9 weeks in non-exercise groups (CON, HFD and HP). For more familiarization with exercise training and testing, exercise groups (HT and HTP) ran on the treadmill (Model T510E, Diagnostic and Research, Taoyuan, Taiwan) for 15 min/d at the speed of 5 m/min for 1 week before starting the HIIT program. After treadmill familiarization, the VO_2_max was measured using a calibrated test in which mice began running at a speed of 8 m per minute with a 5° slope and the treadmill speed gradually increased by 1.8 m/min every 2 min until exhaustion. The aerobic capacity of animals in terms of Vo_2_max was obtained based on the relation of Vo_2_max with speed and treadmill slope. Exercise training was performed for 5 days a week for 8 weeks. Each training session consisted of 30 min running (6 min at the intensity of 50% for a warm-up and 6 min at the intensity of 50 to 60% for a cool down and 18 min for main training). Main training exercise included 3 sets of running at the velocity of 15 m/min (85 to 90% of VO_2_max) with 5° of slope for 4 min with 2 min intervals of slow running (at the intensity of 50 to 60% as active interval rest). After every 2 weeks of exercise, VO_2_max was measured again to determine the intensity of exercise [19].

No electrical shock was employed in order to reduce the effect of stress during the training sessions. To ensure that all exercise mice were running, if the mice stopped running, they were gently pushed forward with the trainers’ hands and forced to continue running. Behavioral tests were performed and body weight was measured under dim red lighting (ap-proximately18lx) over a 2 h period initiated 1h after the onset of the dark phase.

### 2.5. Open Field (OF) Test

The Open Field (OF) test is a valid test used to evaluate the level of anxiety (Stanford 2007). The apparatus of the Open Field test consisted of a square wooden box measuring 60 × 60 × 25 cm. The floor of the arena was divided into sixteen equal squares: four in the center and twelve in the periphery [20]. The total distance (total distance traveled in the apparatus) and average speed (average speed during all movements in the apparatus) was measured to evaluate general locomotor activity. Also, the center time (duration of time spent in central square), wall time (duration of time spent in wall square), corner time (duration of time spent in the corner squares), and the number of excrements were measured to evaluate anxiety. The test duration was 5 min [21,22]. All measurements (times and distances) were recorded by an AHD camera 2-MP, Model No: cp-B20M3, Lens: 3.6 mm-3MP (HIKVISION, Hangzhou, China). The apparatus was cleaned with a solution of 10% ethanol between trials to eliminate animal clues.

### 2.6. Elevated Plus Maze (EPM) Test

The day after the OF test, mice were subjected to the validated test of elevated plus maze (EPM) test which were performed similarly to OF test to asses anxiety. The EPM apparatus (a t-shaped apparatus with 4 arms) made of laminated wood consisted of 2 opposing open arms (7 × 40 cm) and 2 opposing closed arms (7 × 40 cm with 30 cm high walls). The maze was placed 40 cm above the floor. Two of the arms were enclosed by 30 cm tall walls, these are denoted as—closed arms. The other two arms did not have walls, denoted as open arms [23]. All animals were placed in the center of the EPM in the start zone, where the arms intersect, and allowed to explore for five minutes. The percentage of open arm entries (% OAE), as a standard anxiety index, was calculated as follows: % OAE = (entries into the open arms/total entries in any arm) * 100.

The lower ratio is related to the higher anxiety in the mice. Total closed arm entries were measured as a relatively pure index of locomotor activity [24]. All measurements were recorded by an AHD camera 2-MP, Model No: cp-B20M3, Lens: 3.6 mm-3MP. The apparatus was cleaned with a solution of 10% ethanol between trials to eliminate animal clues.

### 2.7. Blood Sampling and Corticosterone Analyses

Blood samples were taken after all of the behavioral tests. Fasted overnight animals were anesthetized by intraperitoneal injection of ketamine (37.5 mg/kg) and xylazine (12.5 mg/kg) mixture. Blood samples were collected into a syringe via cardiac puncture. Samples then were centrifuged at 4000 rpm at 4 °C for 10 min and serum was collected, serum samples were separated and stored at −80 °C prior to assay. Commercially mouse-specific enzyme-linked immunosorbent assay (ELISA) kits were used to determine serum corticosterone level (Cat. No. E-2724, ZellBio GmbH, Lonsee, Germany). The analysis was conducted according to the manufacturer’s instructions. Values were calculated according to a standard curve generated during the experiment [25]. After sacrifying animals, the whole intra-abdominal adipose were removed and weighed (to the nearest 0.001 g) by a digital balance (model GX-400; A&D Company, Tokyo, Japan) for visceral fat mass (VFM).

### 2.8. Statistical Analysis

Statistical analyses were conducted by SPSS^®^ 25.0 (IBM Corporation, Armonk, NY, USA). The normality of data was assessed by Shapiro-Wilk test. Regarding the normal distribution, data were analyzed using one-way analysis of variance (ANOVA) and in the case of significant findings, LSD post hoc test was performed for comparing pairs of means. The statistical significance level was set at *p* < 0.05. Cohen’s d test was used to determine effect size.

## 3. Results

### 3.1. Body Weight and Visceral Fat Mass

Considering body weight, the ANOVA test indicated a significant difference between the study groups following 8 weeks of the high-fat diet [F (4,24) = 26.05, *p* < 0.001]. Comparing paired groups indicated the significant increase of weight in HFD groups compared to the C group (*p* < 0.001).

Following 8 weeks of the main interventions (exercise training, probiotic supplementation, and a combination of them) body weight of five groups compared by ANOVA test which indicated that there was a significant difference between the study groups [F (4,24) = 33.75, *p* < 0.001]. As indicated in Figure 1A, comparing paired groups using the LSD test indicated that body weight was highest in HFD (43.81 ± 2.31), compared to all groups of HTP (29.05 ± 2.92), HT (28.80 ± 1.14), HP (35.00 ± 5.56) and CON (24.20 ± 0.43). Bodyweight of the HTP group was lowest among the HFD groups. Bodyweight of HT mice decreased compared to CON (*p* = 0.022), HFD group (*p* < 0.001), and HP group (*p* = 0.002). However, there was no significant difference between HT and HTP groups (*p* = 0.89).

Probiotic supplementation in the HP group, decreased bodyweight, compared to the HF group (*p* < 0.001). However, the weight of the HP group was higher than HT (*p* = 0.002), HP (*p* = 0.003), and CON (*p* < 0.001) groups. Body weight of combined HIIT and probiotic group reduced significantly compared to HF (*p* < 0.001) and HP (*p* = 0.003) groups. However, the weight of the HTP group was still higher than the CON group (*p* = 0.017).

Figure 1B shows the VFM of C57BL/6 mice in each group at the end of the experiment. A comparison of the five groups indicated a significant difference between groups [F (4,24) = 60.29, *p* < 0.001). As Figure 1B shows, the HFD regime increases the VFM of HFD groups, compared to CON group. VFM of HFD group (1.13 ± 0.12) was highest compared to the other 4 groups of CON (0.16 ± 0.016), HT (0.41 ± 0.05), HP (0.59 ± 0.09), and HTP (0.19 ± 0.06) (*p* < 0.001). This finding was consistent with the result of bodyweight (Figure 1A). In addition, the HFD mice which were subjected to HIIT and probiotics (HTP) had the lowest VFM, compared to all groups with a high-fat diet (*p* < 0.001), while there was no significant difference between HTP and CON groups (*p* = 0.59). VFM in the HT group was lower, compared to the HP group (*p* < 0.001) and HF group (*p* < 0.001), while VFM in the HT group higher, compared to the CON group (*p* < 0.001) and HTP group (*p* < 0.001).

### 3.2. Serum Corticosterone Levels

Comparison of corticosterone in the five groups of the study indicated a significant difference between the groups [F (4,24) = 18.23, *p* < 0.001]. As indicated in Figure 2, Comparing pairs of groups using LSD indicated that corticosterone was highest in the HFD group (262.83 ± 48.39) compared to other four groups including CON (165.10 ± 15.26, d = 2.72), HP (147.38 ± 23.96, d = 3.02), HT (151.65 ± 33.76, d = 2.66), and HTP (132.62 ± 13.43, d = 3.66) (*p* < 0.001). However, there was no significant difference between other study groups (*p* > 0.05). Comparison of HFD and HP groups indicated that probiotics reduced corticosterone levels (*p* < 0.001, d = 3.02).

### 3.3. Anxiety-Related Behavior Test

#### 3.3.1. Open Field Measures

There was no significant difference in wall time between the study groups [F (4,24) = 2.26, *p* = 0.092)] (Table 1; Figure 3A). Wall time was lower in the HFD group (98.20 ± 43.37) compared to CON (139.16 ± 21.7, d = 1.19; *p* = 0.047), HP (160.08 ± 33.32, d = 1.60; *p* = 0.003) and HTP (158.30 ± 18.38, d = 1.80; *p* = 0.004). However, regarding wall time, there was no significant difference between HFD and HT (134.07 ± 29.40, d = 0.96; *p* = 0.066). Also, there was no significant difference between wall times of other study groups (*p* > 0.05). Probiotics consumption in HP compared to HFD groups increased wall time (d = 1.60).

Regarding total distance, a significant difference was found between the study groups [F (4,24) = 68.10, *p* < 0.001] (Table 1). Comparing pairs of means indicated that total distance was highest in HTP (870.31 ± 89.00) compared to other groups including CON (222.43 ± 98.64, d = 6.89), HFD (82.10 ± 33.20, d = 11.73), HP (284.30 ± 70.78, d = 7.28), and HT (475.05 ± 130.40, d = 3.54; *p* < 0.001) and lowest in HFD compared to CON (d = 1.90), HP (d = 3.65), HT (d = 4.12), and HTP (d = 11.37; *p* < 0.001). Total distance in HT group was higher than HP (d = 1.81) (*p* = 0.001), HFD (d = 4.12; *p* < 0.001), and CON (d = 2.18; *p* < 0.001) groups and was lower than HTP group (d = 3.54; *p* < 0.001). Also, total distance of HP group was higher than lower than HTP (d = 7.28; *p* < 0.001) and HT (d = 1.81; *p* = 0.001) groups. There was no significant difference between HP and CON groups (*p* = 0.267). Comparison of HP and HFD groups indicated to probiotics consumption increased total distance (d = 3.65) (Figure 3B).

As regards th corner time, a significant difference was found between the study groups [F (4,24) = 4.22, *p* = 0.010] (Table 1). Comparing pairs of means indicated that corner time was highest in the HFD group (248.68 ± 56.95) compared to other groups of HTP (123.00 ± 53.10, d = 2.28; *p* = 0.001), HT (151.27 ± 83.53, d = 1.36; *p* = 0.007), HP (168.83 ± 44.34, d = 1.56; *p* = 0.24) and CON (144.50 ± 31.42, d = 2.26; *p* = 0.006). Corner time was lower in HTP compared to the HFD group (d = 2.28; *p* = 0.001). There was no significant difference between other study groups (*p* > 0.05). Probiotics consumption decreased corner time in HP compared to HFD groups (d = 1.68) (Figure 3C).

The comparison of the overall speed demonstrated that there was a significant difference between the study groups [F (4,24) = 42.83, *p* < 0.001)] (Table 1). Paired group comparisons indicated that speed was highest in the HTP group (6.35 ± 0.67) compared to all other groups including CON (3.32 ± 1.26, d = 3.00), HFD (1.39 ± 0.44, d = 8.75), HP (3.71 ± 0.36, d = 4.90) and HT (4.56 ± 0.36, d = 3.32) (*p* < 0.001) and lowest in HFD compared to CON (d = 2.24), HP (d = 5.77), HT (d = 7.88) and HTP (d = 8.75) groups (*p* < 0.001). Speed of the HT group was higher than HFD (d = 7.88; *p* < 0.001), HP (d = 2.36; *p* = 0.041), and CON (d = 1.33; *p* = 0.006) groups. Overall speed of the HP group was higher than HFD (d = 5.77) (*p* < 0.001) and lower than HTP (d = 4.90; *p* < 0.001) and HT (d = 2.36; *p* = 0.041) groups. There was no significant difference between HP and CON group (*p* = 0.343). Probiotics consumption increased overall speed in HP compared to HFD groups (d = 5.77) (Figure 3D).

There was a significant difference between center time in the study groups [F (4,24) = 5.24, *p* = 0.002)] (Table 1). Comparing pairs of means indicated that the center time in the HTP group (6.98 ± 0.89) was higher than other groups of HFD (3.62 ± 1.50, d = 2.72; *p* < 0.001), HT (4.54 ± 0.92, d = 2.69; *p* = 0.005) and CON (4.36 ± 1.17, d = 2.52; *p* = 0.004). Probiotics consumption caused the center time of the HP group be significantly higher than the HFD group (5.88 ± 1.96 vs. 3.63 ± 1.50, d = 1.29; *p* = 0.008). There was no significant difference between other study groups. (*p* > 0.05) (Figure 3E).

#### 3.3.2. Elevated Plus Maze (EPM) Test

Percentage of Open Arm Entries (% OAE) in the elevated plus maze test was significantly varied between the five groups [F (4,24) = 41.97, *p* < 0.001] (Table 1). Post-hoc analyses for comparing paired groups (Figure 4) indicated that %OAE of the HTP group (57.58 ± 7.96) was significantly more than all groups including CON (25.90 ± 12.12, d = 3.08), HFT (2.26 ± 3.46, d = 9.01), HP (19.05 ± 7.06, d = 5.12) and HT(30.58 ± 5.86, d = 3.86; *p* < 0.001). Exercise increased %OA in HT group compared to HFD (d = 5.88, *p* < 0.001) and HP (d = 1.77, *p* = 0.015) groups. There was no significant difference between HT and CON groups (*p* > 0.05). Probiotics consumption caused an increase of %OAE of the HP group just compared to the HFD group (d = 3.02), while there was no significant difference between the HP and CON groups (*p* > 0.05).

## 4. Discussion

In the present study, we compared the effect of exercise and probiotic, separately and in combination, in high-fat diet obese mice. We observed that HFD feeding increased body weight and VFM, in C57BL/6 mice. In other words, bodyweight and VFM were highest in the HFD group. A combination of exercise and probiotic reduced VFM, as it was less in the HTP and HT groups respectively compared to other groups. Regarding body weight, although a combination of exercise and probiotic reduced body weight, it was similar to the exercise group and higher than the CON group. In summary, it can be concluded that a combination of exercise and probiotic was more effective than exercise or probiotic separately, in reducing VFM, and exercise was more effective than probiotics in reducing VFM. It seems that VFM is associated with anxiety or depressive symptoms more strongly than body weight [3]. As the findings of the present study indicated, most of the anxiety indices according to open field measures (speed, corner time and distance) and EPM tests (PILA and %OAE) as well as corticosterone were the lowest in the HTP group and highest in HFD group (high-fat diet without probiotic or exercise intervention). A comparison of exercise and probiotic indicated that exercise was more effective than probiotics in reducing most anxiety indices which is in accordance with VFM findings.

In the present experiment, we observed that HFD mice indicated elevated anxiety symptoms according to OF test and EPM test (Figure 3 and Figure 4) and these findings are consistent with previous studies which demonstrated that the HFD regime induces anxiety in rodents [4]. For instance, HFD rats showed significantly greater levels of anxiety-related behavior according to the Light Dark and Open Field tasks [16]. However, some studies observed different findings and reported that the HFD regime has anxiolytic or anxiety-reducing effects in animals [26,27]. The reason for these conflict findings may be due to the duration of HFD exposure. Anxiety levels could be affected by the HFD regime in a bidirectional manner, in which acute exposure to an HFD reduced anxiety levels while chronic access to an HFD increased anxiety levels [4]. In the present study, the HFD feeding continued for 18 weeks and increased most of the anxiety behaviors in mice (Figure 3 and Figure 4). Also, the degree of lipid saturation may affect depressive-like behavior and gut microbiota [28], and may be another cause of discrepancies. The increasing effect of HFD on the anxiety-related behavior in open field task and EPM test were accompanied by increased serum corticosterone levels (Figure 2). There is inconsistency in findings regarding corticosterone response to HFD regimes. Some have reported that the HFD regime increases circulating corticosterone [29], whereas others have found a decrease or no significant influence on circulating corticosterone [30]. It was indicated that there is a close association between and anxiety levels and symptoms of obesity [4], although its exact mechanism(s) is not clear yet. In HFD induced obese animals, resistance to metabolic hormonal signals, such as leptin and insulin may explain the corresponding increases in anxiety levels. In addition, obesity can be related to HPA dysfunctions and increased circulating corticosterone levels which this elevation cause hippocampus impairments, promotion of eating, and fat deposition [31], and all of them may induce the corresponding increases of anxiety levels in HFD animals. So, the HFD regime itself may not directly increase anxiety-related behaviors, but through increasing body weight and fat mass promotes metabolic impairment which is associated with anxiety-related behaviors [4]. However, the relationship between metabolic impairment and anxiety is unclear and its biological mechanisms remain to be understood.

We also observed that some anxiety-related behaviors, bodyweight and VFM decreased in high intensity interval trained mice with HFD regime. Exercise training is considered an effective tool for attenuating metabolic impairment associated with HFD and it can counteract to HFD induced responses of the HPA axis [7]. Available findings have revealed that exercise training reduced chronic corticosterone and limit its action on behavioral responses [32]. However, there is inconsistency in findings. Some studies have reported that exercise training increased circulating corticosterone [33] and another study has demonstrated decreased corticosterone levels by wheel or treadmill running [34]. The conflict of results may arise from mixed interaction of diet and exercise and complex interaction of the neuroendocrine system. In a non-fatty diet, exercise training may lead to increased corticosterone to higher than normal level in order to deal with the exercise energy requirement while in a high-fat diet, exercise training may reduce HPA axis hyperactivity and regulate the circulating corticosterone level to a physiological range [7]. The effects of exercise training on anxiety also have some contradictions in both human and animal studies. Study findings show that exercise training may decrease [35], increase [36], or have no [37] effect on anxiety according to the open field, light-dark box, or elevated-plus maze test. However, a growing body of evidence demonstrated that exercise can alleviate some, but not all, negative cognitive symptoms associated with HFD intake [38]. For instance, 4 weeks of voluntary wheel-running significantly altered behavioral responses to the open field tests [38]. Controversies in findings may be due to the different types of exercise used, as some studies which reported increased anxiety in exercising animals, used forced exercise on a motorized running-wheel which imposes psychological stress on the animals [38].

The main findings of our study revealed that probiotic and HIIT, independently or in combination decreased anxiety- behavior and corticosterone, while the effect of its combination was much more prominent (Figure 3 and Figure 4). Nonetheless, based on available studies, the effect of HIIT on the anxiety of HFD animals has not been explored yet and the present study is the first to investigate this issue. However, the effects of HIIT has been compared with mild-intensity endurance training (ME), combined with a high-fat diet (HFD) on corticosterone levels in rats and observed that exercise training counteract HFD-induced deleterious metabolic function and hypertensive responses of the HPA axis, and HIIT was more effective than ME in the modification of HFD-induced disorders [7].

HFD regime can be associated with increased body fat and marked changes in composition and diversity of gut microbiota. This change can mediate many effects of HFD state and is associated with risks of developing anxiety [39]. Hence, improvement of gut microbiota by probiotics may be an effective intervention for reducing stress and anxiety in HFD feeding subjects [40]. In line with this assumption, we observed that using LG2055 in HFD C57BL/6 mice decreased the anxiety related behavior, serum corticosterone level and VFM. This finding is consistent with previous studies which demonstrated that PS have a positive impact on the management of psychological disorders and animals that received probiotics treatment exhibited anxiolytic-like behavior [11]. Previous study confirmed that probiotics has strong reducing influence on anxiety-like behaviors [11]. Consistent with our observation, a research revealed that the reducing anxiety effect of probiotic supplement was associated with decreased levels of cortisol, fat mass and bodyweight [41]. Probiotics administration to maternally separated rat pups, decreased the serum corticosterone levels [42]. Probiotics successfully suppress the stress response in the HPA axis of adult male mice [43].

Regarding weight modification effects of PS, a reduction in body fat mass and fat percentage was found following probiotic supplementation in pre-obese adults [44]. Also, a high dose of probiotics mixture led to the reduction of fat mass, subcutaneous fat, fat percentage, total cholesterol, and triglycerides in obese postmenopausal women [45]. It appears that probiotics decrease body weight and fat mass probably through several mechanisms including modulation of gut microbiota, decreasing insulin resistance, and increasing satiety [12]. However, some studies have reported different findings. For instance, no change was found in anthropometric indices of obese adolescents with administration of probiotics [46]. Surprisingly, there are reports of increased fat mass in human [47] and weight gain in animal by using probiotic supplements [48]. Possible explanations for these increased weight or fat mass is the ability of some probiotic strains to improve the absorption of nutrients and process in the gut [49] which should be taken into account when comparing research results. Some specific strains of probiotic supplements especially *Lactobacillus rhamnosus GG* has an anti-obesity weight-loss effect [12]. A recent study indicated that probiotic supplementation is a valuable treatment in anxiety and obesity which was accompanied by reducing pro- neuroinflammatory factors [50].

Our findings also demonstrated a synergistic effect of HIIT and probiotic supplementation to reduce anxiety-related factors of VFM, OF task (overall speed, total distance, and center time) and EPM test (overall speed, total distance) and corticosterone. So, it can be concluded that a combination of HIIT and probiotic is a more effective tool for attenuating complications of the HFD regime related to anxiety in C57BL/6 mice.

Considering the available studies, no research was found to examine the effect of exercise training and probiotic supplementation on HFD mice and the present study is regarded the first one which investigated this effect. However, exercise with probiotic treatments are effective in improving spatial memory in Alzheimer’s disease in transgenic mice [50]. HIIT and probiotics may attenuate the metabolic and neuroendocrine disorders induced by the HFD regime which can be associated with elevated anxiety, corticosterone, and body fat levels. However, further studies are needed to determine the mechanisms governing the complex and bidirectional relationship between HFD feeding, HIIT, and probiotics.

Despite the encouraging results, the following limitations should be considered. First, several mediating neuroendocrine factors were not assessed; such factors might have biased two or more measurements in the same or opposite directions. Second, it remains unclear, if and to what extent the present findings are transferable to human non-clinical and clinical samples. Given this, future studies might assess also endocrine factors in both human non-clinical and clinical samples.

## 5. Conclusions

We observed that chronic high-fat diet increased body weight, VFM, serum corticosterone levels, bodyweight, and anxiety-like behaviors and probiotic and HIIT, independently or in combination, attenuated these abnormalities. The combination of probiotic and exercise had synergistic effects on reducing anxiety indices and obesity in high-fat diet mice. For more clarifications, future studies considering HIIT specially combined with probiotics can be suggested for clinical research on patients suffering from anxiety associated with HFD.

## Figures and Tables

**Figure 1 nutrients-13-01762-f001:**
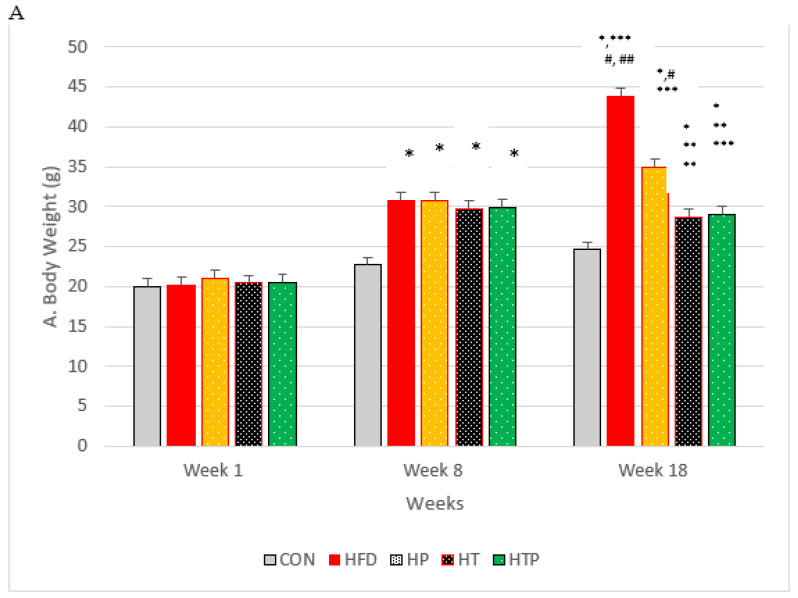
(**A**): Bodyweight of C57BL/6 mice (Mean ± SE) in experimental group, Week 1: first weights, Week 8: following 8 weeks of interventions (high- fat diet in 4 groups and normal diet in 1 group), Week 18: following 8 weeks of interventions (HIIT training and/or probiotic supplementation). (**B**): Comparison of the VFM of C57BL/6 mice (Mean± SE) in experimental groups. * *p* < 0.05: significant difference with CON group; ** *p*: significant difference with HFD; *** significant difference with HP group; #: significant difference with HT group; ##: significant difference with HTP group; CON: control, HFD: high-fat diet, HP: high-fat diet+ probiotic, HT: high-fat diet +HIIT training, HTP: high-fat diet+ HIIT training+ probiotic.

**Figure 2 nutrients-13-01762-f002:**
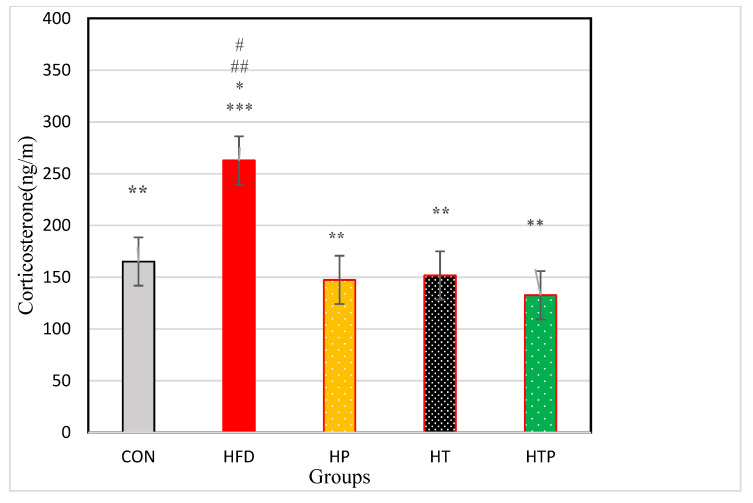
Comparison of the VFM of C57BL/6 mice (Mean ± SE) in experimental groups. * *p* < 0.05: significant difference with CON group; ** *p*: significant difference with HFD; *** significant difference with HP group; #: significant difference with HT group; ##: significant difference with HTP group; CON: control, HFD: high-fat diet, HP: high-fat diet+ probiotic, HT: high-fat diet +HIIT training, HTP: high-fat diet+ HIIT training+ probiotic.

**Figure 3 nutrients-13-01762-f003:**
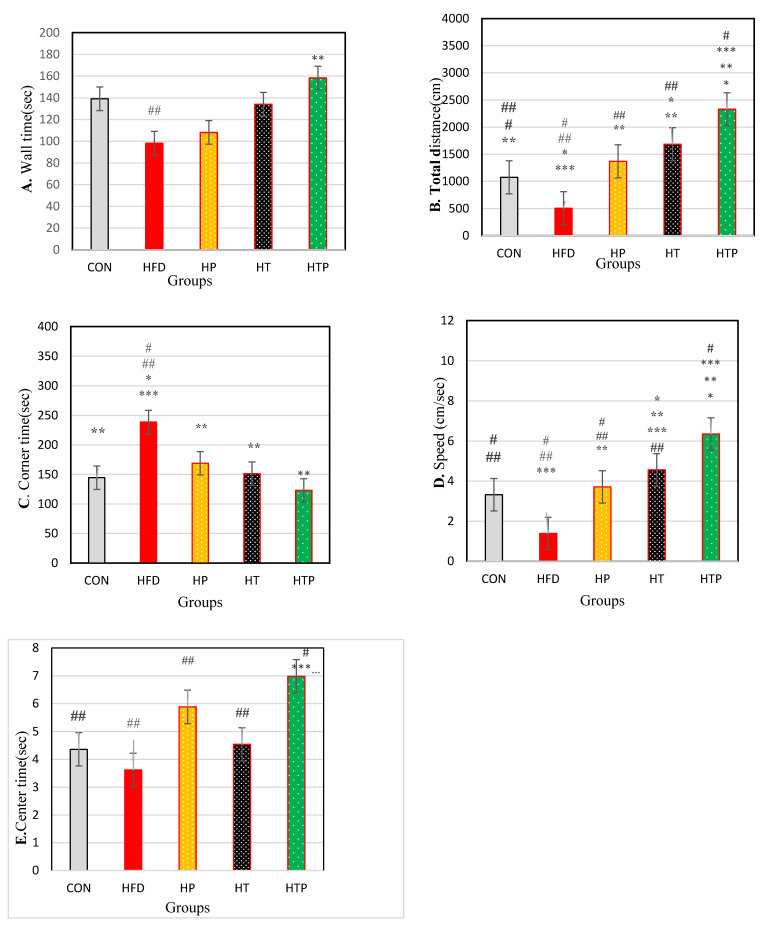
Comparison of the results of the OF tests (Mean ± SE) (**A**): wall time, (**B**): Distance, (**C**): corner time, (**D**): Speed, (**E**): center time in the study groups * *p* < 0.05: significant difference with the CON group; ** *p*: significant difference with the HFD group; *** significant difference with the HP group; #: significant difference with the HT group; ##: significant difference with the HTP group; CON: control, HFD: high-fat diet, HP: high-fat diet + probiotic, HT: high-fat diet + HIIT training, HTP: high-fat diet + HIIT training + probiotic.

**Figure 4 nutrients-13-01762-f004:**
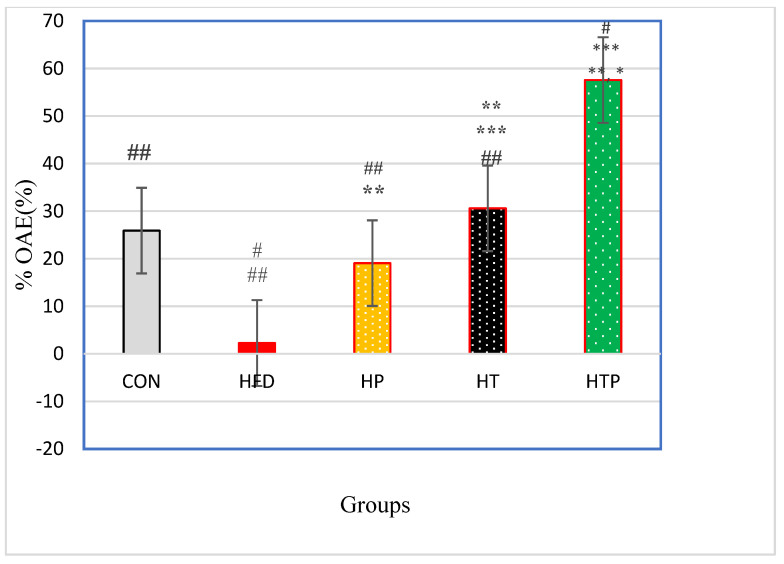
Comparison of %OAE (Mean ± SE) in the experimental groups. * *p* < 0.05: significant difference with the CON group; ** *p*: significant difference with the HFD group; *** significant difference with the HP group; #: significant difference with the HT group; ##: significant difference with the HTP group; CON: control, HFD: high-fat diet, HP: high-fat diet + probiotic, HT: high-fat diet + HIIT training, HTP: high-fat diet + HIIT training + probiotic. OAE: open arm entries.

**Table 1 nutrients-13-01762-t001:** Comparison of variables between the groups of study using one way ANOVA tests.

	Groups					Statistics	
	Con (Mean ± SD)	HFD (Mean ± SD)	HP (Mean ± SD)	HT (Mean±SD)	HTP (Mean ± SD)	F	*p*
Corticosterone (ng/m)	165.10 ± 15.26	262.83 ± 48.39	147.38 ± 23.96	151.65 ± 33.76	132.62 ± 13.43	18.23	<0.001
Wall time (s)	139.16 ± 21.7	98.20 ± 43.37	160.08 ± 33.32	134.07 ± 29.40	158.30 ± 18.38	2.26	0.092
Total distance (s)	222.43 ± 98.64	82.10 ± 33.20	284.30 ± 70.78	475.05 ± 130.40	870.31 ± 89.00	68.10	<0.001
Corner time (s)	144.50 ± 31.42	248.68 ± 56.95	168.83 ± 44.34	151.27 ± 83.53	123.00 ± 53.10	4.22	0.010
Overall speed (cm/s)	3.32 ± 1.26	1.39 ± 0.44	3.71 ± 0.36	4.56 ± 0.36	6.35 ± 0.67	42.83	<0.001
Center time (s)	4.36 ± 1.17	3.62 ± 1.50	5.88 ± 1.96	4.54 ± 0.92	6.98 ± 0.89	5.24	0.002
% OAE (%)	25.90 ± 12.12	2.26 ± 3.46	19.05 ± 7.06	30.58 ± 5.86	57.58 ± 7.96	41.97	<0.001

CON: control, HFD: high-fat diet, HP: high-fat diet + probiotic, HT: high-fat diet + HIIT training, HTP: high-fat diet + HIIT training + probiotic, OAE: open arm entries, SD: standard deviation.

## Data Availability

The data presented in this study are under ownership of the funding institute (Shiraz University) and would not be available.

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
