# Peer review of "Probiotic Supplementation and High-Intensity Interval Training Modify Anxiety-Like Behaviors and Corticosterone in High-Fat Diet-Induced Obesity Mice"

_nutrients, 2021, doi:10.3390/nu13061762_

Round 1

Reviewer 1 Report

The global prevalence of both affective and cognitive disorders is on the rise. Diet has been recognised as one of the leading and at the same time modifiable risk factors. Thus, dietary factors and diet-associated factors should be further investigated in the context of the pathophysiology of these disorders. Taken the previous into account, I found this study novel and of a high value to the ongoing research.

However, I encourage authors to addressed below listed (minor) comments in order to improve the readability of the manuscript.

Introduction

Line 39-41 please not that when considering humans, not HFD but unhealthy dietary practices were demonstrated to be associated with cognitive and affective disorders, including anxiety (PMID: 32340112). Please rephrase.

Line 42 shouldn’t it be “HFD increases”?, please check. I would add also that influence brain vascular system.

Line 48 please note error in the reference editing

Line 58 authors should consider rephrasing the sentence by adding the note that gut microbiota may also influence brain health though so called gut-brain axis (PMID: 29671359).

Line 59 as above, there might be an error in the reference editing (and line 72, 73, 74, 364, 365)

Methods

Was the sample size calculated prior the study?

Results

Legend Figure 4 please indicate for what stands “OAE”

Table 1: what are the units on included numerical values? How is expressed the value mean+/-sd? Please correct. In the table footnote please add the statistical test name used to estimate P value.

Figure 4 I don’t know whether the legend on the right side of the figure is necessary given that the columns have their own legend.

General comments

There are several misspellings along the manuscript.

Editing of the references should be revised.

Discussion

It would be interesting implementing the results form recent study: PMID: 31645150.

Discussion/Conclusions

What could be a possible translation of the obtained results into clinical research?

Author Response

We thank Reviewer 1 for the care devoted to thoroughly review the present manuscript. The Reviewer’s comments were very helpful to improve the quality of the manuscript. Please find the detailed point-by-point-response attached as a separate file.

Again, thank you very much for all your kind efforts.

Reviewer 2 Report

In the nutrients-1231104 titled “Probiotic supplementation and high-intensity interval training modify anxiety-like behaviors and corticosterone in high-fat diet-induced obesity mice” by Parisa Foroozan and colleagues, they have reported that high-intensity interval training and probiotic supplements separately or above all in combination, may have beneficial effects in reducing obesity and anxiety indices. I have few concerns regarding the present manuscript.

-The manuscript is an interesting topic in the field, the introduction is well-organized and written, my first question is my first issue is that some references are in a different format (lines 48, lines 56, for example).

-How the author has calculated the sample size to confirm their results in the present manuscript

- Reading the introduction, my first thought that the authors have measured the gut microbiota, however, this topic is missing in the present manuscript

-The results are interested especially in the group of animals that receive either the high-intensity interval training  and probiotic supplementation, the ANOVA factor is corrected by Bonferroni or another a posteriori test

-The figures need a better format to understand the obtained information

-How the authors have measured the impact of the probiotic administration?

-The authors need to add another analysis to test the efficacy of the treatments in these animals

Author Response

We thank Reviewer 2 for the care devoted to thoroughly review the present manuscript. The Reviewer’s comments were very helpful to improve the quality of the manuscript. Please find the detailed point-by-point-response attached as a separate file.

Again, thank you very much for all your kind efforts.

Round 2

Reviewer 2 Report

Thank you to the authors for taking into account my previous comments about the manuscript, now, the document reads well and no further questions are required